# Revisiting the Integration of Convolution and Attention for Vision Backbone

**Lei Zhu**
City University of Hong Kong
ray.leizhu@outlook.com

**Xinjiang Wang**
Sensetime Research
wangxinjiang@sensetime.com

**Wayne Zhang**
Sensetime Research
wayne.zhang@sensetime.com

**Rynson Lau**[†]
City University of Hong Kong
Rynson.Lau@cityu.edu.hk

## Abstract

Convolutions (Convs) and multi-head self-attentions (MHSAs) are typically considered alternatives to each other for building vision backbones. Although some works try to integrate both, they apply the two operators simultaneously at the finest pixel granularity. With Convs responsible for per-pixel feature extraction already, the question is whether we still need to include the heavy MHSAs at such a fine-grained level. In fact, this is the root cause of the scalability issue w.r.t. the input resolution for vision transformers. To address this important problem, we propose in this work to use MSHAs and Convs in parallel **at different granularity levels** instead. Specifically, in each layer, we use two different ways to represent an image: a fine-grained regular grid and a coarse-grained set of semantic slots. We apply different operations to these two representations: Convs to the grid for local features, and MHSAs to the slots for global features. A pair of fully differentiable soft clustering and dispatching modules is introduced to bridge the grid and set representations, thus enabling local-global fusion. Through extensive experiments on various vision tasks, we empirically verify the potential of the proposed integration scheme, named *GLMix*: by offloading the burden of fine-grained features to light-weight Convs, it is sufficient to use MHSAs in a few (*e.g.*, 64) semantic slots to match the performance of recent state-of-the-art backbones, while being more efficient. Our visualization results also demonstrate that the soft clustering module produces a meaningful semantic grouping effect with only IN1k classification supervision, which may induce better interpretability and inspire new weakly-supervised semantic segmentation approaches. Code will be available at https://github.com/rayleizhu/GLMix.

## 1 Introduction

Since the renaissance of deep learning over a decade ago, CNNs had dominated image analysis, until recently when transformers become popular in vision tasks. CNNs and transformers differ in how they model spatial feature interactions: CNNs use convolutions (Convs), while transformers use multi-head self-attentions (MHSAs). Both have their own advantages and limitations. For example, Convs have an inductive bias of translation equivariance, which matches the image property and enables decent performances with less data. They also have a linear complexity w.r.t. pixel number, making them scalable to high-resolution input. However, they have a limited receptive field, which cannot be remedied simply by stacking more layers together [39]. In contrast, MHSAs can model

---

[†] Rynson Lau is the corresponding author.

38th Conference on Neural Information Processing Systems (NeurIPS 2024).

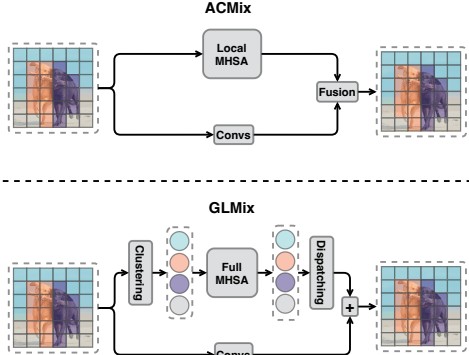

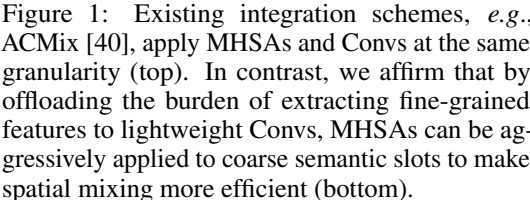

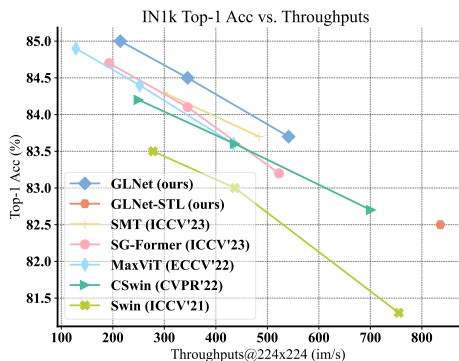

Figure 1: Existing integration schemes, *e.g.*, ACMix [40], apply MHSAs and Convs at the same granularity (top). In contrast, we affirm that by offloading the burden of extracting fine-grained features to lightweight Convs, MHSAs can be aggressively applied to coarse semantic slots to make spatial mixing more efficient (bottom).

Figure 2: IN1k top-1 accuracy vs. FLOPs. While several recent state-of-the-art models (*i.e.*, MaxViT [47], CSWin [14] and SG-Former [43], SMT [34]) lie in almost the same Pareto frontier, our GLNet models move the frontier further to the upper-right with a clear margin.

long-range dependency flexibly, but suffer from a quadratic complexity w.r.t. input resolution and require more data to compensate for the lack of inductive bias. Besides, some discussions [41] also point out that MHSAs play the role of low-pass filters, while Convs play the role high-pass ones. Hence, they are complementary to each other.

There are indeed some works that use both Convs and MHSAs to build vision backbones. Some of them alternate Convs and MHSAs across different stages/blocks [29, 47], forming a loose collaboration. Others [40, 14, 5] integrate Convs and MHSAs tightly in each block. Specifically, they apply Convs and MHSAs in parallel at the same granularity level and fuse their outputs for further processing, as shown in Figure 1(top). With Convs responsible for fine-grained feature extraction, we ask if we still need to apply the heavy MHSAs at the pixel level. Meanwhile, recent vision-language models [1, 63] have shown that an image can be described as a fixed number of visual tokens regardless of its resolution, possibly stemming from the low-rank property of natural signals. Inspired by these works, we propose a global-local mixing (GLMix) block, which uses Convs and MHSAs *at different granularities* for different roles: while Convs focus on extracting local features, MHSAs focus on learning global inter-object relations. Specifically, in each block, we represent an image as both a fine-grained regular grid and a coarse-grained set of semantic slots, and then apply Convs to the grid and MHSAs to the slots in parallel. To enable local-global feature fusion, we introduce a pair of conjugated soft clustering and dispatching modules to bridge the grid and set representations. In this way, we achieve highly efficient local-global modeling by using lightweight Convs to extract high-resolution features and heavy MSHAs to process a fixed number of semantic slots.

To verify the performance of the proposed integration scheme for Convs and MHSAs, referred to as *GLMix*, we start by building a **S**win-**T**iny-**L**ayout model, referred to as GLNet-**STL**, based on the GLMix blocks. GLNet-STL achieves 82.5% top-1 accuracy on ImageNet-1k. It surpasses Swin-T (81.3% top-1 accuracy) significantly by 1.2%. Besides, we note that the macro architectural designs are also important factors for the performance of vision backbones. For example, PoolFormer [64] and ConvNext [36] reveal that with a deeper architecture, vision backbones can still achieve strong performances with simple token mixers such as average pooling and depth-wise convolution. Hence, we further adopt several macro designs from recent state-of-the-art vision backbones [34, 43] and scale the model up to derive a family of 3 models: GLNet-4G/9G/16G. As a result, the GLNet-4G/9G/16G models achieve 83.7%/84.5%/85.0% top-1 accuracy, while being more efficient than recent state-of-the-art works (as shown in Figure 2). Evaluations on downstream dense prediction tasks such as object detection, instance segmentation, and semantic segmentation demonstrate the strengths of GLNet consistently. We also observe that a meaningful semantic grouping effect has emerged in the soft clustering module, even with only image-level classification supervision.

---

Here, we refer to the pixels on the feature maps instead of the input image.

To summarize, our contributions are three-fold:

- We revisit existing integration approaches for Convs and MHSAs, and propose to integrate the two operations *at different granularities*. Such integration combines the strengths of Convs (*e.g.*, the inductive bias of translation equivariance) and MSHAs (*e.g.*, global interactions, data adaptivity) while avoiding the scalability issue w.r.t. the input resolution.
- We introduce a pair of conjugated, fully differentiable clustering and dispatching modules to bridge the set and grid representations of image features, hence enabling the fusion of the global features extracted by MHSAs and local features extracted by Convs. An advantage of the soft clustering module is that it produces meaningful semantic grouping effects without direct dense supervision.
- Through extensive experiments on various computer vision tasks, such as image classification, object detection, and instance/semantic segmentation, we empirically verify our proposed approach. Specifically, a new family of vision backbones, GLNet, demonstrates a favorable performance-computation trade-off to existing state-of-the-arts, under ImageNet-1k supervised training.

## 2 Related Works

**Efficient Attention Mechanisms.** Vanilla MHSAs [48] have a quadratic complexity w.r.t. the number of input tokens, causing huge computation burden and heavy memory footprints, especially in vision applications where the feature maps are in high-resolution. A large volume of works have been conducted to develop efficient variants to overcome such a limitation. These works can be roughly categorized as sparse approximations [6, 35, 70], low-rank approximations [50, 51], and kernel-based methods [27, 7, 19]. The global branch in the proposed GLMix block, which is a combination of soft clustering and dispatching modules and an MHSA, can be used independently as a low-rank attention approximation with not only key-value pairs but also queries being down-sampled. However, according to our experiments, such a usage produces poor performance (Table 6), possibly due to losing too many details and the lack of inductive bias. We find that using MSHAs and Convs in a complementary way is crucial to the success of our proposed GLNet family.

**Hybrid Vision Backbones.** Many works indicate that hybrid vision backbones, which use both Convs and MHSAs, can achieve better performances than pure transformers and CNNs. Among these works, some of them use Convs and MHSAs alternatively across different blocks or stages [34, 29, 47, 59, 12], forming a loose collaboration between the two operators. Another approach adopted by several recent state-of-the-art works [14, 43, 19, 18, 55] is to integrate Convs and MHSAs in each block tightly. Different from these works that apply Convs and MHSAs at the same granularity level, we find that by offloading the burden of extracting fine-grained and location-preserving features to lightweight depth-wise Convs, MHSAs can be applied aggressively on coarse semantic slots while achieving compelling performances with higher efficiency.

**Clustering for Representation Learning.** Clustering is a type of unsupervised learning method used to find meaningful structure, explanatory underlying processes, generative features, and groupings inherent in a set of examples. Existing works, such as [58, 66, 30, 17, 49], have explored clustering for representation learning in deep neural networks. However, unlike ClusTR [58] and TCFormer [66], which use DPC-KNN [16] for the clustering, the soft clustering module in our work is fully learnable and does not rely on predefined rules. In comparison with ClusterFormer [30] and PaCaViT [17], which perform cross-attention between the feature grid and cluster representations/slots, our work performs self-attention over the slots (*i.e.*, queries and key-value pairs are both from the slots), making the attention even more lightweight. Besides, our soft clustering module is hardware-efficient because it is designed to be non-iterative and mainly involves a dense matrix multiplication.

## 3 Methodology

Modern vision backbones are usually built by alternating spatial modeling modules (*e.g.*, Convs, MHSAs, spatial MLPs) and per-location feed-forward networks (FFNs, *i.e.*, embedding MLPs). Much research has been dedicated to developing spatial modeling modules, which is also the primary focus of this work. Specifically, we seek an integration scheme for Convs and MHSAs, which can utilize the strengths of both and scale to high-resolution inputs. Without modifying the standard design of the two basic operators, our key idea is to represent input features twice as both a regular feature grid and a set of *semantic slots*, and then process the feature grid with Convs and the semantic slots with MHSAs (Sec. 3.1). A pair of fully differentiable soft clustering and dispatching modules is

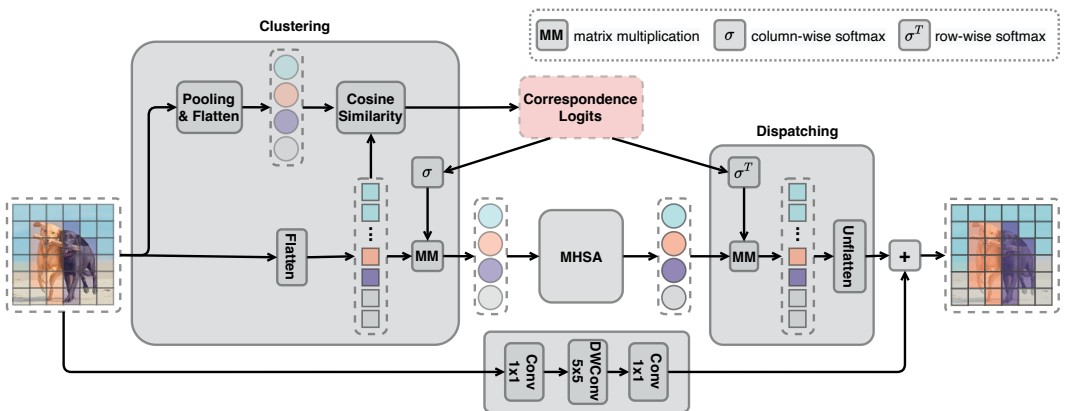

Figure 4: Structure of our GLMix block. At the core is a pair of conjugated soft clustering and dispatching modules to bridge the set and grid representations and enable local-global fusion.

introduced to bridge the two representations, enabling local-global fusion (Sec. 3.2). Based on such an integration scheme we propose a new family of vision backbones named GLNet (Sec. 3.3).

## 3.1 Convs and MHSAs at Different Granularities

Image features are usually organized as a regular grid in vision backbones. Such a representation preserves the spatial correspondence between features and the input image, which is necessary for downstream dense prediction tasks (*e.g.*, semantic segmentation). Besides, extracting local features with the grid representation is convenient and efficient.

In addition to the grid representation, we also create an intermediate set representation composed of a fixed number of *semantic slots* to enable efficient global context modeling. The reason is that although global interactions are usually expensive to compute, it is feasible to use a small amount (*e.g.*, 64 in our experiment) of semantic slots to summarize an image [1, 63], as images are natural signals with heavy spatial redundancy [22]. Notably, the set of semantic slots that we use here is different from the sequence of visual tokens in plain ViTs [15, 46]. While each visual token corresponds to a hard-divided regular patch (*e.g.*, $16 \times 16$ pixels), semantic slots are an abstraction of some "soft" irregular semantic regions, as shown in Figure 3.

We apply Convs to the grid representation to extract local features as they are lightweight and thus efficient in processing the fine-grained feature grid. To model global context, we apply MHSAs on semantic slots. This is a natural choice as MHSAs are permutation-equivariant operators, thus naturally suitable for the set representation. The scalability issue w.r.t. input resolution is avoided, as we have only a small number of semantic slots. Hence, the drawback of MHSAs is overcome.

Next, we illustrate how the set and grid representations are bridged by a pair of soft clustering and dispatching modules, so that local and global features can be fused.

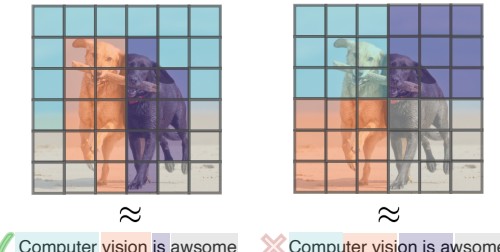

Figure 3: The semantic slots correspond to "soft" irregular semantic regions (left). Compared to using hard-divided regular patches (as adopted by plain ViTs [15, 46]) on the right, our formulation is closer to tokenization in NLP.

## 3.2 Bridging The Set and Grid Representations

To establish a connection between coarse-grained semantic slots and fine-grained feature grids, we need to create a correspondence between them. Although the classical k-means clustering is applicable for this purpose [49, 66], it is suboptimal for two reasons. First, it is an iterative algorithm, which is inefficient on GPUs. Second, it is a heuristic approach, which cannot be end-to-end optimized. Hence,

Table 1: System-to-system comparison with existing works under the **S**win-**T**iny-**L**ayout protocol [70]. †: implemented by us with modules provided by timm [54].

| Models | Params (M) | FLOPs (G) | Throu. (im/s) | IN1K Top-1 (%) |
|---|---|---|---|---|
| Swin-T [35] | 29 | 4.5 | 755.2 | 81.3 |
| DAT-T [56] | 29 | 4.6 | — | 82.0 |
| Swin-ACMix-T [40] | 30 | 4.6 | — | 81.9 |
| Flatten-Swin-T [19] | 29 | 4.5 | — | 82.1 |
| NAT-STL [20] | 29 | 4.5 | — | 81.4 |
| MaxViT-STL† [47] | 28 | 4.5 | 763.4 | 81.8 |
| GLNet-STL (ours) | 30 | 4.4 | 835.9 | **82.5** |

Table 2: Model configurations of the GLNet family. $C$: base channels (*i.e.*, feature channels of the first stage). $e$: FFN expansion ratio. #Blocks: the 4-stage block numbers. FLOPs are measured at $224 \times 224$ resolution.

| Model | $C$ | $e$ | #Blocks | Adv. designs | FLOPs |
|---|---|---|---|---|---|
| GLNet-STL | 96 | 4 | [2, 2, 6, 2] | No | 4.4G |
| GLNet-4G | 64 | 3 | [4, 4, 18, 4] | Yes | 4.5G |
| GLNet-9G | 96 | 3 | [4, 4, 18, 4] | Yes | 9.7G |
| GLNet-16G | 128 | 3 | [4, 4, 18, 4] | Yes | 16.7G |

we introduce a simplified and fully differentiable clustering module and the conjugated dispatching module to address these two problems, as shown in Figure 4. We illustrate the process below.

**Clustering (feature grid → semantic slots).** Given an input feature grid, $\mathbf{X} \in \mathbb{R}^{C \times H \times W}$, where $C$ is the number of channels, $H$ is the height, and $W$ is the width, we first initialize $M$ semantic slots, $\mathbf{S}_{init} \in \mathbb{R}^{M \times C}$, via average pooling followed by shape flattening, as:

$$\mathbf{S}_{init} = \text{Flatten}(\text{AvgPool}(\mathbf{X})). \tag{1}$$

We then compute the correspondence logits, $\mathbf{A} \in \mathbb{R}^{M \times HW}$, as scaled cosine similarity between the initial semantic slots $\mathbf{S}_{init}$ and the flattened input features $\bar{\mathbf{X}} \in \mathbb{R}^{HW \times C}$, as:

$$\mathbf{A} = \text{CosineSimilarity}(\mathbf{S}_{init}, \bar{\mathbf{X}})/\sigma. \tag{2}$$

Here, the learnable scale factor $\sigma$ smooths the distribution of $\mathbf{A}$, preventing dominance by salient entrances. With the correspondence logits, we perform a 1-step update to derive refined semantic slots, $\mathbf{S} \in \mathbb{R}^{M \times C}$, as the weighted sum of flattened features $\bar{\mathbf{X}}$, as:

$$\mathbf{S} = \text{Softmax}(\mathbf{A})\bar{\mathbf{X}}. \tag{3}$$

The refined semantic slots $\mathbf{S}$ are then fed to MHSAs as input.

**Dispatching (semantic slots → feature grid).** After transforming $\mathbf{S}$ to propagate global context with an MHSA module, the transformed semantic slots $\mathbf{S}'$ are dispatched to spatial locations for fusion with local features. Specifically, the dispatched features, $\mathbf{G} \in \mathbb{R}^{C \times H \times W}$, are computed as:

$$\mathbf{G} = \text{Unflatten}(\text{Softmax}(\mathbf{A}^T)\mathbf{S}'). \tag{4}$$

$\mathbf{G}$ can be readily fused with the feature grid processed by Convs due to shape compatibility. We follow the Feature Pyramid Network [32] to use additive fusion, as it is simple/lightweight and provides a regularization that aligns the global and local features in the same semantic space.

**Discussion.** The soft clustering and dispatching operations are highly efficient as the main computations lie in hardware-friendly dense matrix multiplications, and we perform only 1-step instead of iterative updates of the semantic slots. They are fully differentiable as we do not use hard assignments like k-means. The combination is similar to soft-routing in SoftMoE [42], which aims to build large mixture-of-expert models. However, as we target a better balance between cost/performance, our design has several differences: (1) slots are initialized with a different strategy (*i.e.*, per-image average pooling instead of learned parameters shared by all images); (2) the clustering module is placed at a different position (*i.e.*, in pair with token mixers instead of FFNs); and (3) significantly fewer slots are used (*i.e.*, 64, instead of 4096 which is even more than the number of tokens for an image).

## 3.3 GLNet

To verify the performance of the GLMix block proposed above, we start by creating a **S**win-**T**iny-**L**ayout architecture, named GLNet-**STL**, which follows the macro architectural designs of the Swin-T [35] model but with the spatial mixing modules (window attention and shift-window attention) replaced. Specifically, we use the GLMix block in the first three stages of GLNet-STL, where the feature maps are in high-resolution ($\frac{1}{4}$, $\frac{1}{8}$, $\frac{1}{16}$ of the input resolution). At the $4^{th}$ stage, which is

Table 3: Comparison with different state-of-the-art backbones on ImageNet-1K classification. All models are trained and evaluated on $224 \times 224$ input resolution. Top-1 refers to top-1 accuracy (%). We compare models trained with a standard supervised recipe and those trained with an advanced distillation recipe [26].

| Method | | FLOPs | #Param. | Top-1 |
|---|---|---|---|---|
| Standard supervised training | | | | |
| Swin-T [35] | [ICCV'21] | 4.5G | 29M | 81.3 |
| PVT-S [51] | [ICCV'21] | 3.8G | 25M | 79.8 |
| CSWin-T [14] | [CVPR'22] | 4.5G | 23M | 82.7 |
| CMT-S [18] | [CVPR'22] | 4.0G | 25M | 83.5 |
| RegionViT-S [3] | [ICLR'22] | 5.7G | 31M | 83.3 |
| CrossFormer-S [53] | [ICLR'23] | 5.3G | 31M | 82.5 |
| MaxViT-T [47] | [ECCV'22] | 5.6G | 31M | 83.6 |
| MOAT-0 [59] | [ICLR'23] | 5.7G | 28M | 83.3 |
| NAT-T [20] | [CVPR'23] | 4.3G | 28M | 83.2 |
| InternImage-T [52] | [CVPR'23] | 5G | 30M | 83.5 |
| Flatten-T [19] | [ICCV'23] | 4.3G | 21M | 83.1 |
| SG-Former-S [43] | [ICCV'23] | 4.8G | 23M | 83.2 |
| GLNet-4G (ours) | | 4.5G | 27M | **83.7** |
| Swin-S [35] | [ICCV'21] | 8.7G | 50M | 83.0 |
| CSwin-S [14] | [CVPR'22] | 6.9G | 35M | 83.6 |
| RegionViT-M [3] | [ICLR'22] | 7.9G | 42M | 83.4 |
| MaxViT-S [47] | [ECCV'23] | 11.7G | 69M | 84.4 |
| MOAT-1 [59] | [ICLR'23] | 9.1G | 42M | 84.2 |
| NAT-S [20] | [CVPR'23] | 7.8G | 51M | 83.7 |
| InternImage-S [52] | [CVPR'23] | 8G | 50M | 84.2 |
| BiFormer-B [70] | [CVPR'23] | 9.8G | 57M | 84.3 |
| Flatten-S [19] | [ICCV'23] | 6.9G | 35M | 83.8 |
| SG-Former-M [43] | [ICCV'23] | 7.5G | 39M | 84.1 |
| SMT-B [34] | [ICCV'23] | 7.7G | 32M | 84.3 |
| GLNet-9G (ours) | | 9.7G | 61M | **84.5** |

| Method | | FLOPs | #Param. | Top-1 |
|---|---|---|---|---|
| Standard supervised training | | | | |
| Swin-B [35] | [ICCV'21] | 15.4G | 88M | 83.5 |
| CrossFormer-L [53] | [ICLR'22] | 16.1G | 92M | 84.0 |
| CSWin-B [14] | [CVPR'22] | 15.0G | 78M | 84.2 |
| CMT-L [18] | [CVPR'22] | 19.5G | 75M | 84.8 |
| RegionViT-B [3] | [ICLR'22] | 13.6G | 74M | 83.8 |
| MaxViT-B [47] | [ECCV'23] | 23.4G | 120M | 84.9 |
| MOAT-2 [59] | [ICLR'23] | 17.2G | 73M | 84.7 |
| NAT-B [20] | [CVPR'23] | 13.7G | 90M | 84.3 |
| InternImage-B [52] | [CVPR'23] | 16G | 97M | 84.9 |
| Flatten-B [19] | [ICCV'23] | 15.0G | 75M | 84.5 |
| SG-Former-B [43] | [ICCV'23] | 15.6G | 78M | 84.7 |
| GLNet-16G (ours) | | 16.7G | 106M | **85.0** |
| Trained with distillation supervision | | | | |
| Uniformer-S* [29] | [ICLR'22] | 4.2G | 24M | 83.4 |
| Wave-ViT-S* [61] | [ECCV'22] | 4.7G | 23M | 83.9 |
| DualViT-S* [62] | [TPAMI'23] | 5.4G | 25M | 84.1 |
| BiFormer-S* [70] | [CVPR'23] | 4.5G | 26M | 84.3 |
| GLNet-4G* (ours) | | 4.5G | 27M | **84.4** |
| Uniformer-B* [29] | [ICLR'22] | 8.3G | 50M | 85.1 |
| Wave-ViT-B* [61] | [ECCV'22] | 7.2G | 34M | 84.8 |
| DualViT-B* [62] | [TPAMI'23] | 9.3G | 43M | 85.2 |
| BiFormer-B* [70] | [CVPR'23] | 9.8G | 58M | **85.4** |
| GLNet-9G* (ours) | | 9.7G | 61M | 85.3 |

$\frac{1}{32}$ of the input resolution, we use full attention because this is affordable, and beneficial for the performance [29, 41]. As shown in Table 1, our GLNet-STL achieves competitive 82.5% Top-1 accuracy at the highest throughput of 835.9 im/s among several comparable architectures.

The compelling performance of GLNet-STL encourages us to build stronger vision backbones based on it. We therefore investigate several recent state-of-the-arts [14, 47, 70, 43, 34], and incorporate the following advanced architectural designs adopted by them: **(1) Overlapped patch embedding**: use overlapped convolutions ($3 \times 3$ Conv with stride 2) for image/feature down-sampling, instead of non-overlapped ones ($2 \times 2$ Conv with stride 2) as in Swin-Transformer; **(2) Hybrid stage 3**: alternate full MHSAs and GLMix in consecutive blocks of stage 3; **(3) Convolutional position encoding**: add a $3 \times 3$ residual depth-wise convolution prior to each spatial mixing block; **(4) Deeper layout**: increase the depth ([2, 2, 6, 2] $\to$ [4, 4, 18, 4]) while reducing the width (base channel 96 $\to$ 64 and FFN expansion ratio 4 $\to$ 3); and **(5) Convlutional FFN**: add a $3 \times 3$ residual depth-wise convolution between the two linear projections of FFN.

Note that all designs above are widely used in the vision transformers. In addition, as this work is mainly to propose an effective as well as *efficient* integration scheme for MHSAs and Convs, we do not further incorporate some possibly useful designs, such as the squeeze-and-excitation (SE) block [23] used by MaxViT [47] and the gated linear unit (GLU) [44] used by SMT [34]. An ablation study for the incorporated architecture designs can be found in the Supplemental. After applying these modifications sequentially to GLNet-STL, we derive GLNet-4G, a model with 4.5G FLOPs. We scale it up to GLNet-9G and GLNet-16G FLOPs by increasing the number of base channels (96 for GLNet-9G and 128 for GLNet-16G). The model specifications are summarized in Table 2.

## 4 Experiments

We first empirically evaluate our proposed GLNet on a series of computer vision tasks, including ImageNet-1k [13] image classification (Sec. 4.1), COCO [31] object detection and instance segmentation (Sec. 4.2), and ADE20k [69] semantic segmentation (Sec. 4.3). Following existing works, we first train the models for image classification from scratch and then use the trained weights

---

For $224 \times 224$ input image used in IN1k classification, full attention at stage 4 is equivalent to $7 \times 7$ window attention used by Swin-Transformer.

Table 4: Results on the COCO 2017 dataset using the RetinaNet [33] framework for object detection, and Mask R-CNN [21] framework for instance segmentation. $1\times$ refers to 12 epochs, and $3\times$ refers to 36 epochs. MS means multi-scale training. $\text{mAP}^b$ and $\text{mAP}^m$ denote box mAP and mask mAP, respectively. FLOPs are measured at $800 \times 1280$ resolution.

| Backbone | #Param. | FLOPs | RetinaNet 1× | | | | | | RetinaNet 3× + MS | | | | | |
|---|---|---|---|---|---|---|---|---|---|---|---|---|---|---|
| | (M) | (G) | $\text{mAP}^b$ | $\text{AP}^b_{50}$ | $\text{AP}^b_{75}$ | $\text{AP}^b_S$ | $\text{AP}^b_M$ | $\text{AP}^b_L$ | $\text{mAP}^b$ | $\text{AP}^b_{50}$ | $\text{AP}^b_{75}$ | $\text{AP}^b_S$ | $\text{AP}^b_M$ | $\text{AP}^b_L$ |
| PVT-Small [51] | 34 | 226 | 40.4 | 61.3 | 43.0 | 25.0 | 42.9 | 55.7 | 42.2 | 62.7 | 45.0 | 26.2 | 45.2 | 57.2 |
| Swin-T [35] | 39 | 245 | 41.5 | 62.1 | 44.2 | 25.1 | 44.9 | 55.5 | 43.9 | 64.8 | 47.1 | 28.4 | 47.2 | 57.8 |
| Twins-SVT-S [8] | 34 | 210 | 43.0 | 64.2 | 46.3 | 28.0 | 46.4 | 57.5 | 45.6 | 67.1 | 48.6 | 29.8 | 49.3 | 60.0 |
| CrossFormer-S [53] | 41 | 272 | 44.4 | 65.8 | 47.4 | 28.2 | 48.4 | 59.4 | — | — | — | — | — | — |
| MaxViT-T [47] | 46 | 263 | 44.7 | 66.3 | 47.7 | 28.0 | 48.3 | 58.9 | — | — | — | — | — | — |
| BiFormer-S [70] | 35 | 243 | 45.9 | 66.9 | 49.4 | 30.2 | 49.6 | 61.7 | — | — | — | — | — | — |
| SMT-S [34] | 30 | 247 | — | — | — | — | — | — | 47.3 | 67.8 | 50.5 | 32.5 | 51.1 | 62.3 |
| GLNet-4G (ours) | 37 | 214 | **47.1** | **68.6** | **50.5** | **30.8** | **51.1** | **62.9** | **47.9** | **68.8** | **50.8** | **32.7** | **51.6** | **63.5** |
| Swin-S [35] | 60 | 335 | 44.5 | 65.7 | 47.5 | 27.4 | 48.0 | 59.9 | 46.3 | 67.4 | 49.8 | 31.1 | 50.3 | 60.9 |
| Twins-SVT-B [8] | 67 | 326 | 45.3 | 66.7 | 48.1 | 28.5 | 48.9 | 60.6 | 46.9 | 68.0 | 50.2 | 31.7 | 50.3 | 61.8 |
| CrossFormer-B [53] | 62 | 389 | 46.2 | 67.8 | 49.5 | 30.1 | 49.9 | 61.8 | — | — | — | — | — | — |
| ScalableViT-B [60] | 85 | 330 | 45.8 | 67.3 | 49.2 | 29.9 | 49.5 | 61.0 | 48.0 | 69.3 | 51.4 | 32.8 | 51.6 | 62.4 |
| MaxViT-S [47] | 79 | 389 | 46.1 | 68.0 | 49.5 | 28.9 | 50.2 | 61.4 | — | — | — | — | — | — |
| BiFormer-B [70] | 67 | 356 | 47.1 | 68.5 | 50.4 | 31.3 | 50.8 | 62.6 | — | — | — | — | — | — |
| GLNet-9G (ours) | 70 | 292 | **47.7** | **69.0** | **51.6** | **31.8** | **51.6** | **63.5** | **48.8** | **69.6** | **52.5** | **33.5** | **52.9** | **63.9** |

| Backbone | #Param. | FLOPs | Mask R-CNN 1× | | | | | | Mask R-CNN 3× + MS | | | | | |
|---|---|---|---|---|---|---|---|---|---|---|---|---|---|---|
| | (M) | (G) | $\text{mAP}^b$ | $\text{AP}^b_{50}$ | $\text{AP}^b_{75}$ | $\text{mAP}^m$ | $\text{AP}^m_{50}$ | $\text{AP}^m_{75}$ | $\text{mAP}^b$ | $\text{AP}^b_{50}$ | $\text{AP}^b_{75}$ | $\text{mAP}^m$ | $\text{AP}^m_{50}$ | $\text{AP}^m_{75}$ |
| Swin-T [35] | 47.8 | 264 | 42.2 | 64.6 | 46.2 | 39.1 | 61.6 | 42.0 | 46.0 | 68.2 | 50.2 | 41.6 | 65.1 | 44.8 |
| Twins-SVT-S [8] | 44.0 | 248 | 43.4 | 66.0 | 47.3 | 40.3 | 63.2 | 43.4 | 46.8 | 69.2 | 51.2 | 42.6 | 66.3 | 45.8 |
| CSWin-T [14] | 42 | 279 | 46.7 | 68.6 | 51.3 | 42.2 | 65.6 | 45.4 | 49.0 | 70.7 | 53.7 | 43.6 | 67.9 | 46.6 |
| BiFormer-S [70] | 45.2 | 262 | 47.8 | 69.8 | 52.3 | 43.2 | 66.8 | 46.5 | — | — | — | — | — | — |
| SGFormer-S [43] | 41 | — | 47.4 | 69.0 | 52.0 | 42.6 | 65.9 | 46.0 | 49.6 | 71.1 | 54.5 | 44.0 | 68.3 | 46.9 |
| SMT-S [34] | 40.0 | 265 | 47.8 | 69.5 | 52.1 | 43.0 | 66.6 | 46.1 | 49.0 | 70.1 | 53.4 | 43.4 | 67.3 | 46.7 |
| GLNet-4G (ours) | 46.6 | 233 | **48.3** | **70.3** | **53.3** | **43.6** | **67.3** | **46.9** | **49.9** | **71.6** | **54.7** | **44.5** | **68.8** | **48.1** |
| Swin-S [35] | 69.1 | 354 | 44.8 | 66.6 | 48.9 | 40.9 | 63.4 | 44.2 | 48.5 | 70.2 | 53.5 | 43.3 | 67.3 | 46.6 |
| CrossFormer-B [53] | 72 | 408 | 47.2 | 69.9 | 51.8 | 42.7 | 66.6 | 46.2 | — | — | — | — | — | — |
| CSWin-S [14] | 54 | 342 | 47.9 | 70.1 | 52.6 | 43.2 | 67.1 | 46.2 | 50.0 | 71.3 | 54.7 | 44.5 | 68.4 | 47.7 |
| BiFormer-B [70] | 76.3 | 375 | 48.6 | 70.5 | 53.8 | 43.7 | 67.6 | 47.1 | — | — | — | — | — | — |
| SGFormer-M [43] | 51 | — | 48.2 | 70.3 | 53.1 | 43.6 | 66.9 | 47.0 | 50.5 | 71.5 | 54.9 | 45.4 | 68.8 | 48.2 |
| SMT-B [34] | 51.7 | 328 | 49.0 | 70.2 | 53.7 | 44.0 | 67.6 | 47.4 | 49.8 | 71.0 | 54.4 | 44.0 | 68.0 | 47.3 |
| GLNet-9G (ours) | 79.5 | 311 | **49.5** | **71.4** | **54.5** | **44.5** | **68.5** | **48.0** | **51.0** | **72.0** | **56.1** | **46.2** | **69.5** | **48.7** |

for model initialization when performing downstream dense prediction tasks. Note that for dense prediction tasks which take high-resolution inputs, we keep the number of semantic slots to 64, which is consistent with that of image classification. We have found that 64 slots are sufficient to achieve state-of-the-art performances while increasing the number does not help. We then visualize the semantic slots to demonstrate that a meaningful semantic grouping effect emerges in the proposed soft clustering module (Sec. 4.4). Finally, we conduct an ablation study on the design choices of the GLMix integration scheme, which is the core of GLNet (Sec. 4.5).

## 4.1 Image Classification on ImageNet-1k

**Settings.** For a fair comparison with existing works, we conduct image classification experiments on the ImageNet-1k dataset [13], using the standard training recipe provided by Swin-Transformer [35] and the advanced distillation recipe provided by LV-ViT [26]. The training details can be found in Appendix B. We then evaluate the models for classification accuracy and benchmark their throughputs with the script provided by the timm library [54], following the same hardware (a single Tesla V100 32G GPU) and batch size (128) configurations used in Swin-Transformer [35].

**Results.** In Table 3, we compare GLNet with several closely related methods and/or recent state-of-the-arts. Under the setting of standard supervised training, our GLNet-4G/9G/16G consistently show comparable or superior performances to existing best-performing models across different model scales. With dense distillation supervision, the potential of GLNets is further unleashed compared to standard supervised training. For example, the accuracy of the GLNet-4G model increases from 83.7% to 84.4%, a significant performance improvement of 0.7%. Both GLNet-4G and GLNet-9G provide a more competitive performance-FLOPs trade-off than other distilled models. As FLOPs is an indirect metric for practical inference speed and does not consider the memory access cost, we also plot the performance-throughput curve in Figure 2. The improvements become more pronounced when viewed w.r.t. throughputs. Interestingly, several of the latest vision backbones (*i.e.*, SG-Former [43],

MaxiViT [47], and CSwin [14]) lie in almost the same Pareto frontier, while our GLNet models further move the frontier to the upper-right corner with a clear margin. Such a result demonstrates the superiority of our integration philosophy: by applying the heavy MHSAs at a coarse granularity and light-weight Convs at a fine granularity, spatial modeling can be both effective and efficient.

## 4.2 Object Detection and Instance Segmentation

**Settings.** We evaluate the backbones for object detection and instance segmentation on COCO 2017 [31]. All experiments are conducted using the MMDetection [4] toolbox to ensure a fair comparison with existing works. The RetinaNet [33] framework is used for object detection, and the Mask R-CNN [21] framework is used for instance segmentation. During training, we initialize the backbone with weights trained on ImageNet-1K while leaving all other layers randomly initialized. Input images are resized by fixing the shorter side to 800 pixels while restricting the longer side to no more than 1,333 pixels. We train the RetinaNet and Mask R-CNN detectors with $1\times$ schedule (12 epochs) and $3\times$ schedule (36 epochs) provided by MMDetection. More training details are provided in Appendix B. We report the widely used average precision (AP) metric family, such as mean average precision (mAP), average precision at different thresholds ($AP_{75}$ and $AP_{50}$), and average precision for objects of different sizes ($AP_S$, $AP_M$ and $AP_L$). Details of these metrics can be found in MMDetection [4].

**Results.** We show the results for object detection and instance segmentation in Table 4. Our method achieves the best performances among the compared methods across all metrics and the two model sizes in both cases. These results indicate that local-global modeling with the GLMix block benefits object/instance-level tasks.

## 4.3 Semantic Segmentation on ADE20K

**Settings**. Our semantic segmentation experiments are conducted on the ADE20K dataset using the MMSegmentation [10] toolbox. We evaluate our approach using two frameworks - Semantic FPN [28] and UperNet [57]. In both cases, the backbone is initialized with ImageNet-1k weights, while the other layers are randomly initialized. For a fair comparison, we follow the same setting as PVT [51] to train the model 80k steps in our Semantic FPN experiments. On the other hand, for our UperNet experiments, we follow the settings used in Swin Transformer [35] and train the model for 160k iterations. More training details are provided in Appendix B. We report the mean intersection over union (mIoU) metric with no test-time augmentation.

Table 5: Performance comparison of different backbones on the ADE20K segmentation task. We report mIoU with no test-time augmentation. FLOPs are computed at $512 \times 2048$ resolution.

| Backbone | Semantic FPN 80k | | | UperNet 160k | | |
|---|---|---|---|---|---|---|
| | #Param. (M) | FLOPs (G) | mIoU (%) | #Param. (M) | FLOPs (G) | mIoU (%) |
| PVT-S [51] | 28.2 | 161 | 39.8 | — | — | — |
| Swin-T [35] | 31.9 | 182 | 41.5 | 59.9 | 945 | 44.5 |
| Twins-SVT-S [8] | 28.3 | 144 | 43.2 | 54.4 | 901 | 46.2 |
| CSWin-T [43] | 26.1 | 202 | 48.2 | 59.9 | 959 | 49.3 |
| BiFormer-S [70] | 29.3 | 173 | 48.9 | 55.3 | 930 | 49.8 |
| SG-Former-S [43] | 25.4 | 205 | 49.0 | 52.5 | 989 | 49.9 |
| SMT-S [34] | — | — | — | 50.1 | 935 | 49.2 |
| GLNet-4G (ours) | 30.7 | 150 | **49.6** | 56.8 | 907 | **50.6** |
| PVT-M [51] | 48.0 | 219 | 41.6 | — | — | — |
| Swin-S [35] | 53.2 | 274 | 45.2 | 81.3 | 1038 | 47.6 |
| Twins-SVT-B [8] | 60.4 | 261 | 45.3 | 88.5 | 1020 | 47.7 |
| CSWin-S [14] | 38.5 | 271 | 49.2 | 64.6 | 1027 | 50.4 |
| BiFormer-B [70] | 60.4 | 282 | 49.9 | 88.5 | 1041 | 51.0 |
| SG-Former-M [43] | 38.2 | 273 | 50.1 | 68.3 | 1114 | 51.2 |
| SMT-B [34] | — | — | — | 61.8 | 1004 | 49.6 |
| GLNet-9G (ours) | 63.6 | 230 | **51.3** | 91.7 | 988 | **51.4** |

**Results**. Table 5 shows the results of the two different frameworks. Our GLNet-4G/16G achieve 49.6/51.3 mIoU with the Semantic FPN framework, improving the previous best SG-Former-S/M by 0.6/1.2 mIoU. A similar performance gain for the UperNet framework is also observed. The enhancements demonstrate the benefits of utilizing GLNet for high-resolution pixel-wise predictions.

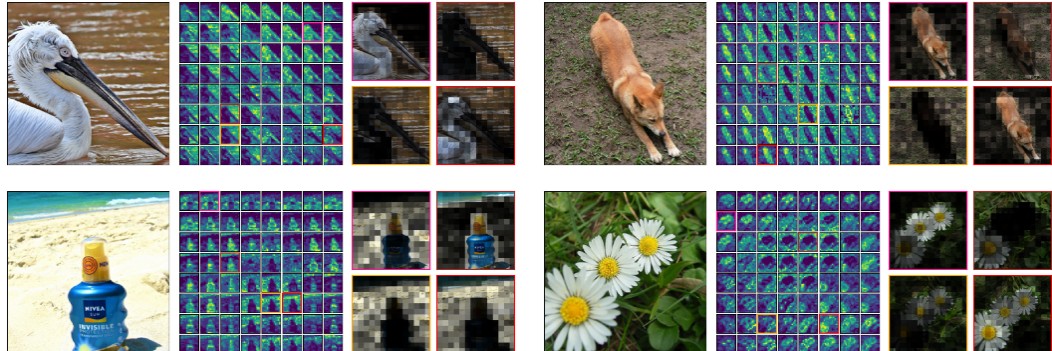

Figure 5: Visualization of semantic slots. For each sample, we show the input image (left), assignment maps of all semantic slots (middle), and four representative slots (right). We use the k-medoids algorithm to select the four representative slots automatically.

## 4.4 Visualization of Semantic Slots

As mentioned in Sec. 3.2, the conjugated clustering and dispatching modules construct a correspondence between the semantic slots and the feature grid. Such a formulation allows us to visualize which regions the semantic slots correspond to. Specifically, we extract the clustering weights in Eq. 3 and split them into M scalar maps of shape H × W. These scalar maps are then pseudo-colored for visualization. In addition, we use the k-medoids algorithm to select four representative slots for a closer look automatically. We find that a meaningful semantic grouping effect emerges in the first block of stage 3, as shown in Figure 5. Note that we use ImageNet-1k trained GLNet-STL for visualization. Hence, the model receives no dense supervision. Visualization for more samples, more blocks and at different epochs can be found in Appendix C.

## 4.5 Ablation Study

We ablate our GLMix integration scheme using the GLNet-STL model. By default, we use a global branch with 64 semantic slots and a local branch with $5 \times 5$ depth-wise conv in parallel in the GLMix blocks, as shown in Figure 4. With this default setting, we investigate the effect of **(a)** local-global collaboration, **(b)** the clustering strategy, **(c)** Conv kernel size in the local branch, and **(d)** number of slots in the global branch. Table 6 shows the experimental results. We summarize our findings below.

**Local-global collaboration.** First, using both local and global branches together is crucial. With the global/local branch removed, the model has a significantly degraded accuracy of 81.8%/78.0%, indicating that both coarse-grained inter-object relationship and fine-grained per-pixel local context are important. Second, using global and local branches in parallel instead of sequentially is important. A possible explanation is that the parallel layout provides a regularization for the global branch from the local branch. Otherwise, the global branch is difficult to optimize due to the lack of inductive bias. Finally, using Convs in the local branch is better than window MHSAs, as the latter are heavier and significantly decrease the throughput from 835.9 im/s to 660.9 im/s. This may be because Convs can implicitly bring position information via padding [25, 9] while window MHSAs cannot.

**Clustering strategy.** The soft clustering approach is an important component of the GLMix block. Using the k-means clustering results do not only produce a significantly lower throughput (835.9 im/s → 440.6 im/s) but also incurs unstable training. This can be attributed to the fact that k-means is an iterative, non-differentiable algorithm, as mentioned in Sec. 3.2. We also observe that initializing the semantic slots as learnable parameters decreases the accuracy from 82.5% to 82.1%. This implies that per-image adaptive initialization is better than static initialization. Possibly, there are difficulties to learn diverse contexts for each image with shared parameters as the slot initialization, according to visualizations in Appendix C.

**Convolution kernel size in the local branch.** The model is robust to the convolution kernel size in the local branch. Using a kernel size of 3 or 7 produces a similar accuracy (82.4%) to the kernel size of 5 (82.5%). This is because the global branch has provided a sufficient large receptive field.

Table 6: Ablation study on the GLMix design choices. We investigate the effect of **(a)** local-global collaboration, **(b)** the clustering strategy, **(c)** convolution kernel size of the local branch, and **(d)** number of slots in the global branch. †: W-MHSA stands for window MHSA; we use a window size of 7 because size divisibility is required. ‡: It is implemented with the official release of Clustered Attention [49], NaN loss occurred during training.

| Model | Slot init. | Slot number | Conv k.s. | FLOPs (G) | Params (M) | Throu. (im/s) | IN1k Top-1 (%) |
|---|---|---|---|---|---|---|---|
| GLNet-STL | pooling | 64 | 5 | 4.4 | 30.3 | 835.9 | 82.5 |
| local branch only | pooling | - | 5 | 3.8 | 26.4 | 999.7 | 81.8 |
| global branch only | pooling | 64 | - | 3.8 | 28.3 | 982.4 | 78.0 |
| sequential (global → local) | pooling | 64 | 5 | 4.4 | 30.3 | 860.1 | 80.6 |
| sequential (local → global) | pooling | 64 | 5 | 4.4 | 30.3 | 825.9 | 79.6 |
| local branch w/ W-MHSA† | pooling | 64 | w7 | 5.0 | 32.2 | 660.9 | 81.1 |
| k-means clustering‡ | hashing | 64 | 5 | 5.2 | 30.3 | 440.6 | N/A |
| static slot initialization | param. | 64 | 5 | 4.4 | 30.5 | 852.0 | 82.1 |
| local w/ 7 × 7 DWConv | pooling | 64 | 7 | 4.4 | 30.3 | 855.2 | 82.4 |
| local w/ 3 × 3 DWConv | pooling | 64 | 3 | 4.3 | 30.4 | 823.9 | 82.4 |
| global w/ 9 slots | pooling | 9 | 5 | 3.9 | 30.3 | 893.6 | 81.9 |
| global w/ 25 slots | pooling | 25 | 5 | 4.0 | 30.3 | 880.8 | 82.1 |
| global w/ 36 slots | pooling | 36 | 5 | 4.1 | 30.3 | 880.0 | 82.3 |
| global w/ 49 slots | pooling | 49 | 5 | 4.2 | 30.3 | 866.6 | 82.3 |
| global w/ 81 slots | pooling | 81 | 5 | 4.5 | 30.3 | 790.0 | 82.4 |

**Number of semantic slots in the global branch.** Using 64 semantic slots is sufficient to achieve a good performance. Although the accuracy decreases to 82.3% with fewer semantic slots (*e.g.*, 9, 25, 36 or 49), increasing the number to 81 also incurs a small performance drop to 82.4%. We hypothesize that this is due to the optimization difficulty caused by too many similar/near-duplicate slots [68].

## 5 Conclusion

In this paper, we have revisited the existing integration approaches for Convs and MHSAs, and proposed to apply the two operators at *different granularity levels*. We discover that by offloading the task of extracting fine-grained features to the lightweight Convs, the heavy MHSAs can be aggressively applied to a few semantic slots. Such an integration scheme, named GLMix, enables highly efficient local-global modeling to build high-performance vision backbones. A key component of GLMix is a pair of conjugated soft clustering and dispatching modules for bridging the feature grid and the set of semantic slots. Meaningful semantic grouping effects, which may induce better interpretability and inspire new weakly-supervised semantic segmentation approaches, are observed in the clustering process.

Currently, we only consider using a static number of semantic slots (*i.e.*, 64 in our experiments) for all images. This may cause many redundant slots representing the same content, as shown in Figure 5. It may be interesting to design a dynamic slot pruning mechanism for more efficient computation and end-to-end weakly-supervised segmentation. Another drawback of GLNet is that it still incorporates many hardware-inefficient depth-wise convolutions with low arithmetic intensity. Seeking more hardware-friendly alternatives will further improve its throughputs on modern hardware.

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

## A  Effect of Advanced Architecture Designs

| Architecture design | Params (M) | FLOPs (G) | Throu. (im/s) | IN1k Top-1 (%) |
|---|---|---|---|---|
| Swin-T layout (GLNet-STL) | 30.3 | 4.4 | 835.9 | 82.5 |
| + overlapped patch emb. | 32.3 | 4.7 | 782.4 | 82.7 |
| + hybrid stage 3 | 31.4 | 4.8 | 784.0 | 83.1 |
| + convolution pos. enc. | 31.4 | 4.8 | 762.6 | 83.2 |
| + deeper layout | 26.8 | 4.5 | 630.7 | 83.5 |
| + conv. FFN (GLNet-4G) | 27.0 | 4.5 | 541.2 | 83.7 |

Table 7: The evolution path from GLNet-STL to GLNet-4G. Modifications are applied sequentially.

As mentioned in Sec. 3.3, we incorporate several advanced architecture designs adopted by recent vision backbones in our GLNet family to achieve state-of-the-art performance. Here, we list the effects of these designs in Table 7. All these designs improve the accuracy. However, they also decrease the throughput.

## B  Training Details

This section provides more training details for ImageNet-1k image classification, COCO object detection and instance segmentation, and ADE20K semantic segmentation.

**Image classification.** For the standard supervised training recipe, training details are in Table 8. When training with the advanced distillation recipe [26], we add an extra distillation head to the GLNet-4G/9G model and use the NFNet-F6 [2] to generate distillation targets; other training details are shown in Table 9. Experiments are run on 16 Tesla V100 SXM2 (32GB) GPUs. Each experiment takes 2-4 days, depending on model size.

**Object detection and instance segmentation.** For COCO experiments, all models are trained using the AdamW [38] optimizer with a batch size of 16. We use a linear schedule with 500 warm-up iterations and set the peak learning rate as $1e-4$. The weight decay is 0.05 for Mask R-CNN [21] and 0.001 for RetinaNet [33]. Experiments are run on 8 or 16 Tesla V100 SXM2 (32GB) GPUs. Each experiment takes 1-2 days, depending on model size.

**Semantic segmentation.** For the ADE20K semantic segmentation task, we apply the AdamW optimizer with a batch size of 32. In Semantic FPN [28] experiments, we use the cosine annealing learning rate schedule with 1000 warm-up iterations and a peak learning rate of 2e-4. The weight decay is $1e-4$. In UperNet [57] experiments, a polynomial learning rate schedule is employed with a linear warm-up phase of 1500 iterations. We set the learning rate as $6e-4$ and weight decay as $1e-2$. Experiments are run on 8 Tesla V100 SXM2 (32GB) GPUs. Each experiment takes 1-2 days, depending on model size.

## C  More Visualization Results

In this section, we provide more visualization results, including (1) visualization of semantic slots of blocks at different depths (Figure 6), (2) visualization of slot evolution over training epochs (Figure 7), and (3) visualization of slots using learned parameters as clustering initialization (Figure 8). We summarize the main observations as below:

- The semantic slots at the lower block ($2^{nd}$ block) tends to group pixels according to color cues. At the middle block ($5^{th}$ block), an object-level grouping effect has emerged. The upper block ($10^{th}$ block) pays attention to discriminative local regions.
- During the training, we found that at the end of the $1^{st}$ epoch we can already distinguish the foreground objects and the backgrounds, although the grouping has not very concentrated patterns, this is possibly due to the fact that even a random projection can preserve distances/similarities well. At the end of the $5^{th}$ epoch, the semantic grouping becomes more concentrated and similar to that of the final stage.

Table 8: Training details of standard supervised training for ImageNet-1k classification.

| config | value |
| --- | --- |
| optimizer | AdamW [38] |
| learning rate | 2e-3 |
| weight decay | 0.05 |
| optimizer momentum | $\beta_1, \beta_2 = 0.9, 0.999$ |
| batch size | 2048 |
| learning rate schedule | cosine decay [37] |
| warmup epochs | 5 |
| training epochs | 300 |
| augmentation | RandAug(9, 0.5) [11] |
| label smoothing [45] | 0.1 |
| mixup [67] | 0.8 |
| cutmix [65] | 1.0 |
| gradient clip | 5.0 |
| drop path [24] | 0.15/0.3/0.4 |

Table 9: Training details of the advanced distillation recipe [26] for ImageNet-1k classfication.

| config | value |
| --- | --- |
| optimizer | AdamW |
| learning rate | 2e-3 |
| weight decay | 0.05 |
| optimizer momentum | $\beta_1, \beta_2 = 0.9, 0.999$ |
| batch size | 2048 |
| learning rate schedule | cosine decay |
| warmup epochs | 5 |
| training epochs | 310 |
| augmentation | RandAug(9, 0.5) |
| label smoothing | 0.1 |
| mixup | 0.8 |
| cutmix | 1.0 |
| gradient clip | 5.0 |
| drop path | 0.1 |

- When the semantic slots are initialized with learned parameters instead of adaptive average pooling as we proposed, the slots are quite more chaotic and tend to focus on foreground objects only, indicating that there are difficulties to learn diverse and global contexts. This possibly accounts for the degraded classification accuracy with such a design (82.5% → 82.1%, Table 6).

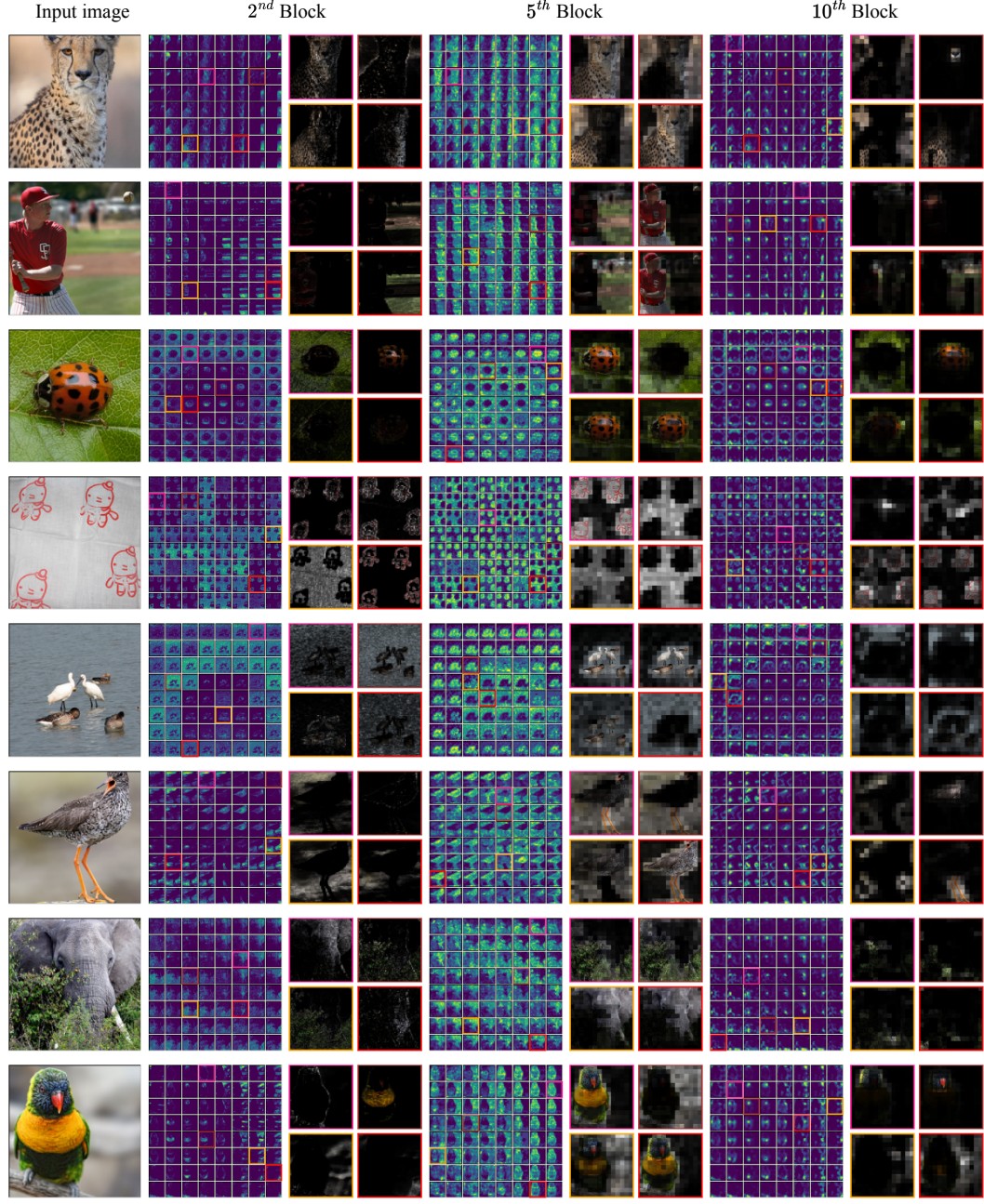

Figure 6: Visualization of semantic slots of blocks at different depths. The setting is the same as in Figure 5, except that we add the visualizations for a shallower block (columns 2-3) and a deeper block (columns 6-7).

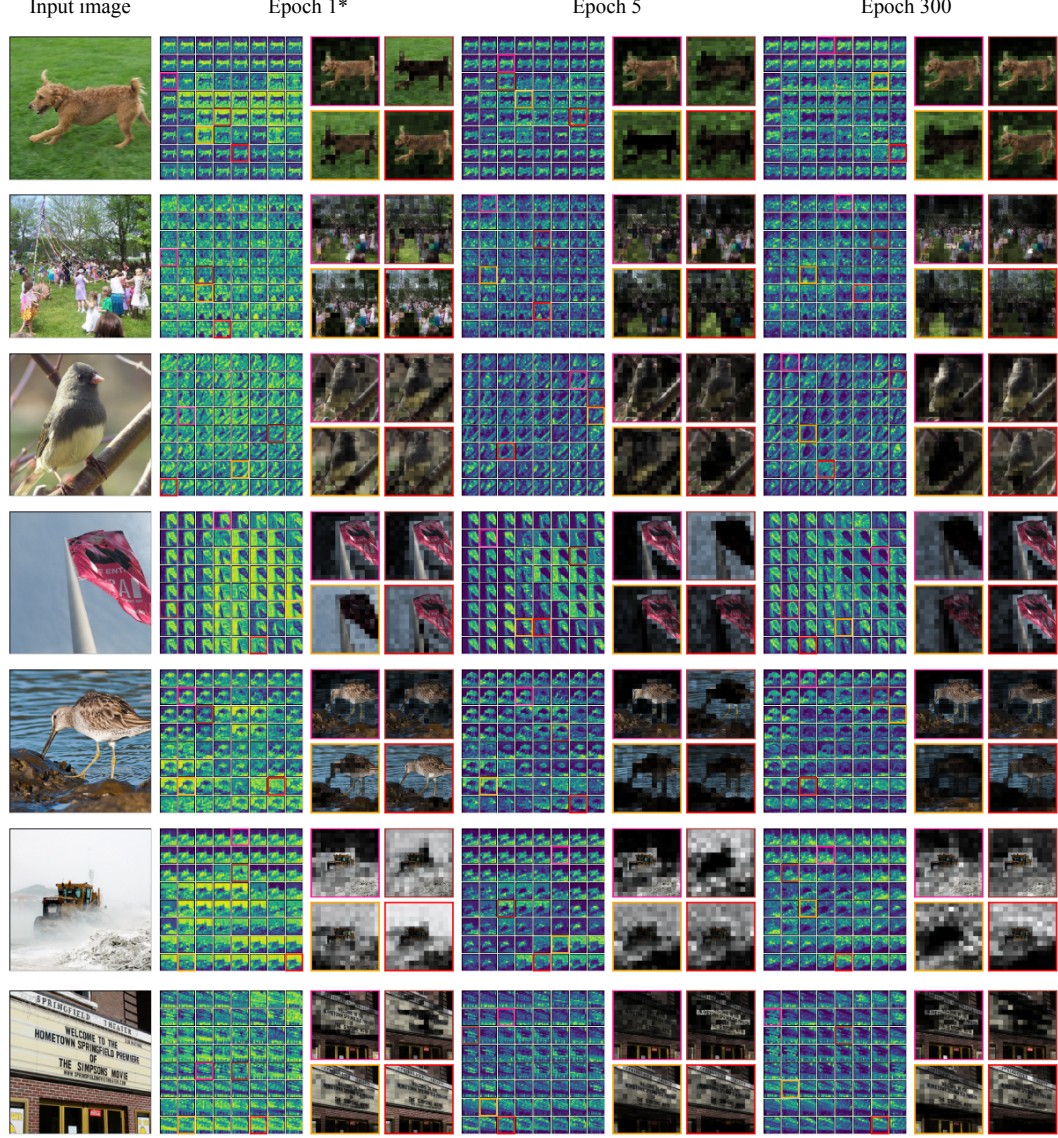

Figure 7: Slot evolution over training epochs. The setting is the same as in Figure 5, except that the checkpoint is replaced. *: slot assignments of epoch 1 are normalized to range $[0, 1]$ for better visibility, otherwise most of them will look like either empty or randomly scattered patterns.

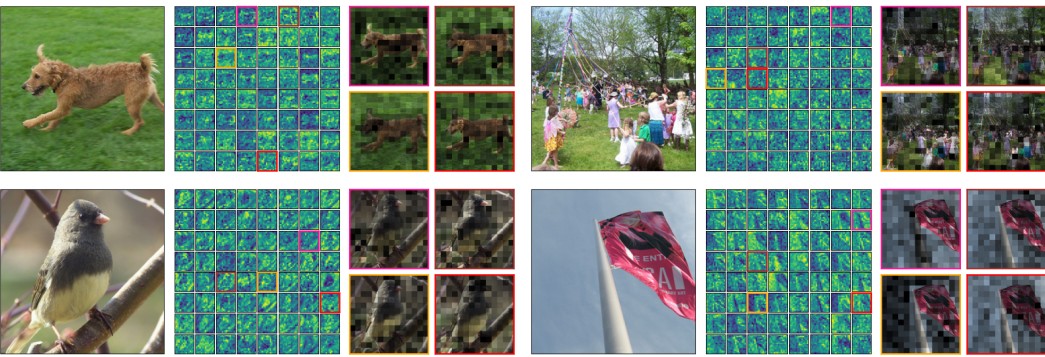

Figure 8: Visualization of slots using learned parameters as clustering initialization. The setting is the same as in Figure 5, except that in the soft clustering initialization is modified.

