# OpenReview forum: "Revisiting the Integration of Convolution and Attention for Vision Backbone"
_NeurIPS.cc/2024/Conference — NeurIPS 2024 poster_

### Official Review · Reviewer_txiT · 2024-07-07

**Soundness:** 3
**Presentation:** 3
**Contribution:** 3
**Rating:** 5
**Confidence:** 5

**Summary:**

This paper addresses the scalability issue in vision transformers by integrating convolutions (Convs) and multi-head self-attentions (MHSAs) at different granularity levels, rather than at the finest pixel level. The authors propose using Convs for fine-grained per-pixel feature extraction and MHSAs for coarse-grained semantic slots in parallel. They introduce a pair of fully differentiable soft clustering and dispatching modules to bridge the grid and set representations, enabling effective local-global fusion. The proposed integration scheme, named GLMix, offloads fine-grained feature extraction to light-weight Convs and utilizes MHSAs in a limited number of semantic slots. Through extensive experiments, they demonstrate the efficiency and effectiveness of their approach, showing improved performance and interpretability in various vision tasks. The paper also highlights the potential of their method to induce better semantic segmentation with weak supervision.

**Strengths:**

1. Innovative Integration Approach: The idea of integrating Convs and MHSAs at different granularity levels is innovative and addresses the scalability issues inherent in vision transformers.

2. Efficient Local-Global Fusion: The use of soft clustering and dispatching modules enables efficient local-global feature fusion, which is a significant advancement.

3. Extensive Empirical Validation: The method achieves state-of-the-art results on ImageNet-1k, COCO, and ADE20K benchmarks, showing a favorable performance-computation trade-off. Furthermore，the semantic grouping effect observed in the clustering module enhances the interpretability of the model.

**Weaknesses:**

1. Complexity of Implementation: The introduction of soft clustering and dispatching modules adds complexity to the implementation, which might pose challenges for practical deployment.

2. Static Number of Semantic Slots: The use of a static number of semantic slots for all images may lead to redundancy and inefficiency in certain cases.

3. Limited Scope of Clustering Strategy: The clustering strategy, though effective, could be further optimized. The current implementation might still be computationally intensive for real-time applications.

**Questions:**

1.How sensitive is the overall performance of the GLMix model to the choice of clustering strategy? Have the authors tested alternative clustering methods, and what were the outcomes?

2.What are the potential limitations of using a static number of semantic slots, and have the authors considered dynamic allocation of semantic slots based on the complexity of the input image?

3.How adaptable is the GLMix integration scheme to different types of vision tasks, such as object detection, instance segmentation, and semantic segmentation? Are there any task-specific adjustments needed for optimal performance?

4.What are the scalability limits of the GLMix models when dealing with very high-resolution images or videos?

**Limitations:**

1. Scalability to Higher Resolutions: While the method shows good performance on standard benchmarks, its scalability to very high-resolution images or videos remains to be fully explored.

2. Hardware Inefficiency: The use of depth-wise convolutions, despite their low arithmetic intensity, might still be inefficient on certain hardware, limiting the practical applicability of the model.

---

> ### Author Rebuttal · Authors · 2024-08-06
>
> W1: Introducing the soft clustering and dispatching modules adds complexity to the implementation, which might pose challenges for practical deployment.
>
> The clustering and dispatching modules involve only standard and widely used operators such as Matrix Multiplication and SoftMax, which are supported by all deep learning libraries across different hardwares, such as CPUs and GPUs. Therefore, the complexity of the implementation should not be a problem.
>
> ---
>
> W2: The use of a static number of semantic slots for all images may lead to redundancy and inefficiency in certain cases.
>
> The static number of semantic slots is indeed a limitation of our method, as we mentioned in Section 5. However,
>
> * On the one hand, we have observed that such a simple strategy is already sufficiently efficient to achieve a good efficiency-performance tradeoff, as demonstrated in Figure 2.
> * On the other hand, dynamic allocation of the slots still has the limitation on efficiency in the typical batched inference scenarios: an image with fewer slots needs slot padding to align an image with more slots.
>
> Although it may not be easily addressed, it would be interesting for future work to consider the dynamic allocation of slots.
>
> ---
>
> W3: The clustering strategy, though effective, could be further optimized. The current implementation might still be computationally intensive for real-time applications.
>
> * Our clustering strategy is designed to be lightweight and only involves a pooling operation, a matrix multiplication followed by SoftMax. The three operations are widely used and highly efficient in all deep learning libraries.
> * The clustering takes less than 5% FLOPs of the whole models. It is far from being a computation bottleneck.
>
> ---
>
> Q1: How sensitive is the overall performance of the GLMix model to the choice of the clustering strategy? Have the authors tested alternative clustering methods, and what were the outcomes?
>
> The design choice of the clustering strategy is significant. According to our ablation study on the clustering strategy in L312-317 and Table 6,
> * using k-means clustering not only produces a significantly lower throughput (835.9 im/s -> 440.6 im/s) but also incurs unstable training;
> * clustering initialization with per-image adaptive pooling improves the performances over static initialization with learnable parameters (82.5% vs. 82.1%).
>
> ---
>
> Q2: What are the potential limitations of using a static number of semantic slots, and have the authors considered the dynamic allocation of semantic slots based on the complexity of the input image?
>
> We have addressed this issue in W2. Please refer to our reply to W2 above.
>
> ---
>
> Q3: How adaptable is the GLMix integration scheme to different types of vision tasks, such as object detection, instance segmentation, and semantic segmentation? Are there any task-specific adjustments needed for optimal performance?
>
> * Keeping the number of slots consistent (i.e., 64) with image classification is good enough for dense prediction tasks. For object detection (Table 4 top group）, instance segmentation (Table 4 bottom group), and semantic segmentation (Table 5), we do not apply any task-specific adjustments and still obtain state-of-the-art performances.
> * We have also tried using more slots in the GLNet-4G backbone for semantic segmentation with UperNet. Using more slots, such as 100 or 256, did not improve the performances (50.6 mIoU -> 50.5 mIOU or 50.6 mIOU).
>
> ---
> Q4: What are the scalability limits of the GLMix models when dealing with very high-resolution images or videos?
>
> * With GLMix, memory and computation grow linearly w.r.t. the input size, as the clustering and dispatching take linear complexity, and the attention among slots takes constant complexity. Therefore, the scalability should be similar to linear attention models such as SwinTransformer.
> * In practice, the scalability limits depend on memory and latency constraints, as well as engineering optimizations. For reference, below we show the peak memory occupancy and latency w.r.t. different input resolutions and batch sizes. Note that the single-input cases (batch size = 1) are heavily unoptimized in Pytorch and can be further improved with advanced inference engines like Nvidia Triton.
>
> | Batch Size | Input Size | GLNet-STL | GLNet-4G | GLNet-9G | GLNet-16G |
> |---|---|---|---|---|---|
> |128|$224^2$|1957MB/149ms|1553MB/233ms|2373MB/392ms|3230MB/586ms|
> |128|$448^2$|7469MB/562ms|5889MB/1012ms|8762MB/1690ms|11682MB/2517ms|
> |1  |$896^2$|346MB/22ms|433MB/59ms|722MB/95ms|1057MB/137ms|
> |1  |$1792^2$|1108MB/108ms|5027MB/628ms|7599MB/970ms|10218MB/1329ms|

---

> > ### Comment · Reviewer_txiT · 2024-08-12
> > **Response to authors**
> >
> > I've thoroughly reviewed the authors' responses and appreciate their thoughtful engagement. I will stay in touch for further discussion as we approach the final rating.

---

> ### Author Response · Authors · 2024-08-13
>
> Thanks for your response. We hope our previous reply has addressed your main concerns. Please feel free to reach out if you have any additional questions or require further clarification on any aspects of our work.

---

### Official Review · Reviewer_CazW · 2024-07-09

**Soundness:** 3
**Presentation:** 2
**Contribution:** 2
**Rating:** 4
**Confidence:** 4

**Summary:**

The authors propose to leverage the strengths of convolution layers and MHSA block to improve the performance of vision transformers. They propose to apply them in parallel at different granularity, such that the convolution layers are applied to the grid of local features and MHSA to slots for global features. Slots can be considered as clustering of patch features, where patches corresponding to the same object would be associated to a slot. To create slots, they use soft clustering that enables local-global fusion across patch features such that it produces some meaningful semantic grouping effect.
To connect the slots back to the patch features, they propose a dispatching module, which is then fused with a convolution block os 1x1 convolution and depth-wise convolution operation (similar to Convnext). They terms this entire block as GLMix for Global-local mixing block.

Finally, to validate the effectiveness of their approach, the authors propose steps to adapt it to Swin-tiny-layout module and empirically show the performance for image classification on Imagenet-1k, semantic segmentation on ADE20K, object detection and instance segmentation on MS-COCO. The authors provide a comprehensive list of baseline methods (Table 3, Table 4) for different tasks and also provide visualization of the slots attending to different regions of the image. They also discuss different experimental settings and provide additional visualizations of the slots in the Appendix.

**Strengths:**

The following are the strengths of the paper:
- The authors propose an idea to integrate the strengths of convolution i.e. ability to encode inductive bias, alog with MHSA block with ability to learn global representation, to improve vision transformers. The idea is quite interesting and while the use of convolution and transformer layers have been applied in Hybrid architectures, their use in parallel is under-studied and is an exciting topic.
- The visualizations provided in Figure 5 of the object level representations learned by the slots and additional such visualizations in the Appendix, gives a nice insight into representations encoded by slots. Its also interesting to see the slots learning different regions of the image, not only focusing on the object, but also regions in the background that might be of interest.
- I also really liked the through and extensive experiments, along with comparisons to different SoTA methods, not only in terms of performance but also in efficiency.
- The paper is also easy to understand, with exhaustive mentions of related works and their comparisons.

**Weaknesses:**

I have the following concerns about the work:

- In L59, the authors state that they have applied several macro designs from SMT [32]. However, comparing the performance of SMT, SMT-S achieves similar performance as GLNet-4G i.e. 83.7% accuracy, but has much fewer parameters (20.5M vs 27M), while having similar FLOPs. Additionally, the variant SMT-B has 2G fewer flops and half the number of parameters and yet achieves the same performance  i.e 84.3% s. 84.5%. Furthermore, SMT-S is also better than GLNet-4G on object detection on MS-COCO, which using RetinaNet. The authors of SMT also provide visualizations of the attention heads clearly attending to the object, same as in GLNet. This means that the scale-aware-module (SAM) achieves a much more efficient integration of convolution with MHSA. Could the authors please comment on effectiveness of GLNet against SMT.
- Maybe I missed it, but Im unable to understand the intuition of applying self-attention operation on the slots (L161). Instead of just having a cosine similarity between the slots and patch features as in eq(2), a softmax over the slots would ensure that the slots do not collapse to the same object. Is this the reason why authors apply MHSA on S to obtain S'?
- Continuing from the above point, I further cannot understand the significance of the dispatching module. Could the authors please explain the intuition behind going from slots to patch features? Is it because, it would be easier to integrate with the features obtained after conv layers?
- In Figure 4, the clustering before applying the GLMix block and after are exactly the same. Its not clear to me what is being learnt by the GLMix block.
- A popular work that uses slots to perform object centric learning is DINOSAUR [1*], which show in Figure 6 of their paper that increasing the number of slots results in a drop in performance. The authors however find that varying the slots has almost no effect in performance. Could the authors please comment of this difference in observation? Additionally, by carefully looking in figure 5, we can observe that there are many slots that represent the same object. This leads to some redundancy in learning. Can the authors vary the slots to somewhere between 5-10 like the one performed in [1*] to see if they have the same observation?

[1*] Seitzer et al., Bridging the Gap to Real-World Object-Centric Learning, ICLR 2023

**Questions:**

Apart from the ones mentioned in the Weakness section, I have the following questions

- Can the authors please clarify their architecture? I see references to Swin, ConvNext, LV-ViT, etc. But Im unsure about what a single layer of GLNet looks like? Im also curious if the authors use a pretrained backbone, because Im unsure about the performance gains with just 1 V100 GPU and training the slots over just 1 iteration. Maybe Im missing some detail here.
- Is there any specific reason apart from the smaller feature resolution that the GLMix block is applied to the 3rd layer?
- There are missing references to hybrid ViTs, which the authors can add and discuss.

[2*] Venkataramanan et al., Skip-Attention: Improving Vision Transformers by Paying Less Attention, ICLR 2024
[3*] Pan et al., Less is more: Pay less attention in vision transformers., AAAI 2022
[4*] Mehta & Rastegari, Separable self-attention for mobile vision transformers, arxiv  2022
[5*] Graham et al.,  Levit: a vision transformer in convnet’s clothing for faster inference, ICCV 2021

- Could the authors also provide the throughput details for table 3,4 and 5

**Limitations:**

Yes the authors have discussed the limitation, but one can reduce the redundancy in slots by trying to empirically understand the performance of the proposed method with fewer slots, as suggested above.

---

> ### Author Rebuttal · Authors · 2024-08-06
>
> W1: Please comment on the effectiveness of GLNet against SMT.
>
> * For classification, the throughput/efficiency of SMT is not as good as its parameters and FLOPs indicate. This is because its core design, scale-aware modulation (SAM), relies heavily on depthwise convolutions (DWConvs), which cannot utilize the GPUs well due to low arithmetic intensity. Note that although we also use DWConvs, we only use it once in each GLMix block.
>
> | Model |#Params(M)|FLOPs(G)|Acc.@IN1k(%)|Throu.(im/s)|
> |---|---|---|---|---|
> |SMT-S|20.5|4.7|**83.7**|484|
> |GLNet-4G|27|4.5|**83.7**|**541**|
> | | | | | |
> |SMT-B|32.0|7.7|84.3|298|
> |GLNet-9G|61|9.7|**84.5**|**345**|
>
> * For object detection (Table 4) and semantic segmentation (Table 5), GLNet performs better and has lower FLOPs. This implies that GLNet has better scalability to high-resolution inputs than SMT.
>
> |Backbone|#Params(M)|FLOPs(G)|mAP (RetinaNet 3X+MS @ COCO) |
> |---|---|---|---|
> |SMT-S|30|247|47.3|
> |GLNet-4G|37|**214**|**47.9**|
>
> |Backbone|#Params(M)|FLOPs(G)|mIoU (UperNet 160k @ ADE20K) |
> |---|---|---|---|
> |SMT-S|50.1|935|49.2|
> |GLNet-4G|56.8|**907**|**50.6**|
> | | | | | |
> |SMT-B|61.8|1004|49.6|
> |GLNet-4G|91.7|**988**|**51.4**|
>
> ---
> W2: What's the intuition of applying self-attention operation on the slots? Is it to ensure that slots do not collapse to the same object?
>
> * The self-attention operation on the slots is for inter-object/inter-region relation learning.
> * We have tried removing the self-attention over slots, and the slots do not collapse to the same object, possibly because the column-wise and row-wise SoftMax operations (Eq. 3 and Eq. 4) have introduced both inter-slot and inter-position competitions. However, the IN1k accuracy drops from 82.5% to 82.0% for the GLNet-STL.
>
> ---
> W3: Could the authors please explain the intuition behind the dispatching module? Is it because it would be easier to integrate with the features obtained after conv layers?
>
> The goal of the dispatching module is to propagate inter-region/object relation to spatial positions in the feature grid. Indeed, it is crucial for the fusion with the Conv layer processed features, as you point out here.
>
> ---
> W4: In Figure 4, the clustering before and after applying the GLMix block are exactly the same. It's not clear what is being learnt by the GLMix block.
>
> The GLMix block performs feature transformation. The input and output of the GLMix block are both feature maps. In Figure 4, we colorize both the input and output feature maps according to the expected clustering so that they look the same, which may be the reason for the confusion. We will revise the figure to be more clear (e.g., by removing the clustering colorization on the input feature map). Thanks for pointing out the issue.
>
> ---
> W5: The observation on the effects of varying number of slots differs from DINOSAUR. Could the authors please comment on this difference? Can the authors vary the slots to somewhere between 5-10?
>
> * There are at least two aspects that make our observation on slot numbers differ from DINOSAUR:
>     * Supervision signal. DINOSAUR is an unsupervised framework for object discovery while our GLNet is trained under a supervised learning framework. DINOSAUR may require setting a proper number of slots to serve as a prior on how many objects can emerge in an image from their datasets.
>     * Evaluation metric. DINOSAUR uses the foreground adjusted rand index (FG-ARI), a no-reference metric for clustering quality, while we use the IN1k accuracy in the ablation study.
> * We have tried to use 9 slots (initialized with 3 $\times$ 3 pooling) in GLNet-STL; the IN1k accuracy drops from 82.5% to 81.9%.
>
> ---
> Q1: Can the authors please clarify their architecture? What does a single layer look like in GLNet? Do the authors use a pretrained backbone?
>
> The architecture design is specified in Section 3.3. To summarize,
> * The macro architecture of GLNet follows most existing hierarchical transformers: it has four stages with gradually smaller spatial resolutions (1/4, 1/8, 1/16, 1/32) and larger hidden dimensions (C, 2C, 4C, 8C). Each layer contains a spatial mixing block (the GLMix block or MHSA block) followed by a channel mixing FFN/MLP. The detailed configurations can be found in Table 2.
> * We do not use a pretrained backbone. Since there are architectural differences, it is impossible to inherit the weights from existing architectures. We train the model from scratch on ImageNet and then use the weights to initialize the backbone for downstream dense prediction tasks, following existing works such as SwinTransformer.
>
> ---
> Q2: Is there any specific reason apart from the smaller feature resolution that the GLMix block is applied to the 3rd layer?
>
> The GLMix block is applied to the 1st, 2nd, and 3rd stages, and each stage has several layers according to the model size (see Table 2). The 4th stage uses a standard attention block as the resolution (1/32 input size) is sufficiently small so that the standard attention is affordable and beneficial for the performance. See also Section 3.3 for more details.
>
> ---
> Q3: There are missing references to hybrid ViTs, which the authors can add and discuss.
>
> GLNet distinguishes itself from existing works, including the recommended references, by applying MHSAs and Convs at different granularity levels, i.e., the object/region level and pixel level. Thanks for bringing these works to our attention. We will reference and discuss them in our revision.
>
> ---
> Q4: Could the authors also provide the throughput details for table 3,4 and 5.
>
> * It isn't easy to provide throughputs for these tables because most of the compared works did not report the data.
> * For classification (Table 3), Figure 2, together with the SMT throughputs in W1, has covered the latest SOTA models for comparison.
> * For object detection (Table 4) and semantic segmentation (Table 5), the throughputs may not be that useful because the task-specific heads, instead of the backbone, consume most computations.

---

> > ### Comment · Reviewer_CazW · 2024-08-09
> > **Response to author's rebuttal**
> >
> > Thanks to the authors for their response and pointing out the results for object detection and semantic segmentation. The authors have clarified some of my queries. The following are some points, which are still unclear to me.
> >
> > - With regard to the GLMix block, my main query about intuition was trying to understand more about what each component is learning. To me, the architecture can be compressed into 3 components:
> >     - slots to patch feature cross attention
> >     - self-attention of output of above step to obtain refined slots
> >     - patch features to refined slot cross attention, which is then combined with output of convolution features
> >
> > If this is the case, like i mentioned before, I am curious to see the visualization on images on ImageNet-1k, where objects are not object-centric unlike the ones shown in Figure 5. This would help better understand what different slots observe and whether they form nice clusters. I understand as per NeurIPS policy, the authors can upload a 1 page document containing figures and tables.
> > - Following up with the above point, the authors mention that they do not use a pre-trained model. In this case, during the initial phase of the training, the slots would be random features. Im curious to know how does it affect the learning? Can the authors show any visualization of how the slots evolve and as function of #epochs? More importantly, Im curious to understand how these random slots is not detrimental to the overall learning, as it would form incorrect object clusters.
> >
> > - Instead of making the slots conditioned on the input image, Im curious to know what happens when the slots are made independent similar to Perceiver.
> >
> >
> >
> > Im open to discussing this with the authors in case of any misunderstanding from my end.
> > Thanks

---

> ### Author Response · Authors · 2024-08-10
> **Clarification on the GLMix block, random slots during early epochs, and additional visualizations**
>
> Thanks for the response. Below are our replies to the new comments.
>
> ---
>
> C1: further clarification on the GLMix block.
>
> Your latest comments on the GLMix block are mostly correct. However, we would like to clarify one thing. Although the clustering and dispatching modules are similar to cross attention, the two modules are designed to be lightweight so that they have **no QKV projections**, **no multihead mechanism**, and **the two modules share the correspondence/assignment logits**.
>
> ---
>
> C2: the slots are random features during early epochs. Does this affect the learning?
>
> We did not encounter any difficulties caused by the random slots in early epochs. Previous Perceiver/DETR-like architectures, which involve something similar to the slots, also have no problems here. One possible reason for this phenomenon is that even random projection can preserve distances/similarities well.
>
> ---
>
> C3: request for the visualization of non-object-centric images from IN1k, of semantic slots evolution against training epochs, and of semantic slots with Perceiver-like non-image-conditioned design.
>
> * Unfortunately, while authors could optionally upload a PDF in the global response section during the initial rebuttal phase, we are not allowed to edit this section now. Therefore, we are unable to submit a PDF now.
> * By looking at the visualization of non-object-centric images, we still observe meaningful semantic grouping effects of the things and stuff.
> * Unfortunately, we did not keep the model snapshots at different epochs or for the Perceiver-like non-image-conditioned design, which are required to visualize the latter two items. We will redo the experiments and will describe the results as soon as possible. However, since we are attending the ACL conference now, it may take a couple of days for us to get back to you, but we will try our best to reply soon.

---

> ### Author Response · Authors · 2024-08-13
> **Update on the visualization**
>
> We have done some further visualizations as per your request. Below are our observations.
>
> * At the end of the 1st epoch, a few semantic slots can already be associated with foreground objects, while the others are either uniform or scattered patterns. At the end of the 5th epoch, the semantic grouping effect becomes similar to Figure 5 in the paper. In subsequent epochs (e.g., 10th, 50th, ...), the grouping effect gradually becomes more obvious.
> * With non-image-conditioned initialization for the slots, while a similar semantic grouping effect is observed, we find that nearly all slots are associated with the foreground objects in many cases. This possibly accounts for the degraded classification accuracy with such a design (82.5% -> 82.1%, Table 6): the background information is ignored, even though it may be helpful in some cases.

---

### Official Review · Reviewer_fhiH · 2024-07-14

**Soundness:** 3
**Presentation:** 3
**Contribution:** 3
**Rating:** 5
**Confidence:** 5

**Summary:**

The paper discusses the use of Convolutions (Convs) and multi-head self-attentions (MHSAs) in vision backbones. Traditionally considered alternatives, the authors question the need for both to operate at the finest pixel granularity, particularly highlighting the scalability issues this causes in vision transformers. They propose a novel approach where Convs and MHSAs work in parallel but at different granularities: Convs handle local features on a fine-grained grid, while MHSAs manage global features on a coarse-grained set of semantic slots. The integration is facilitated by soft clustering and dispatching modules that bridge these representations, enabling local-global fusion. Their method, GLMix, leverages lightweight Convs for fine details and uses MHSAs on fewer semantic slots.

**Strengths:**

1. The paper re-evaluates current methods of integrating Convs and MHSAs and proposes a novel integration at different granularities. This approach harnesses the advantages of Convs, such as translation equivariance, and MHSAs, such as global interactions and data adaptivity, while mitigating scalability issues related to input resolution.
2. The paper presents a pair of fully differentiable clustering and dispatching modules that connect the set and grid representations of image features. This enables the fusion of global features from MHSAs and local features from Convs. A notable advantage of the soft clustering module is its ability to produce meaningful semantic groupings without requiring direct dense supervision.

**Weaknesses:**

1. In Line 6, this paper mentions "the scalability issue," but it only contains experiments on the IN-1K size dataset. Such a dataset cannot evaluate "scalability." I suggest that the authors revise this argument.
2. The novelty that introduces clustering into attention is limited. [1,2,3,4] have already studied the clustering in attention. Please compare with these methods and highlight your novelty and contributions.

[1] Xie, Y., Zhang, J., Xia, Y., Hengel, A. V. D., & Wu, Q. (2022). Clustr: Exploring efficient self-attention via clustering for vision transformers. arXiv preprint arXiv:2208.13138.
[2] Liang, J., Cui, Y., Wang, Q., Geng, T., Wang, W., & Liu, D. (2024). Clusterfomer: clustering as a universal visual learner. Advances in neural information processing systems, 36.
[3] Grainger, R., Paniagua, T., Song, X., Cuntoor, N., Lee, M. W., & Wu, T. (2023). PaCa-ViT: learning patch-to-cluster attention in vision transformers. In Proceedings of the IEEE/CVF Conference on Computer Vision and Pattern Recognition (pp. 18568-18578).
[4] Zeng, W., Jin, S., Liu, W., Qian, C., Luo, P., Ouyang, W., & Wang, X. (2022). Not all tokens are equal: Human-centric visual analysis via token clustering transformer. In Proceedings of the IEEE/CVF Conference on Computer Vision and Pattern Recognition (pp. 11101-11111).

**Questions:**

Please refer to the weakness.

**Limitations:**

This paper will address the limitations in the conclusion section.

---

> ### Author Rebuttal · Authors · 2024-08-06
>
> W1: IN-1k size dataset cannot evaluate "scalability." It is advisable to revise this argument.
>
> We thank this reviewer for the suggestion. However, in Line 6, we actually refer to the scalability w.r.t. the input size, instead of the dataset size. This can be reflected by the FLOPs-performance tradeoff on dense prediction tasks such as semantic segmentation (Table 5) and object detection (Table 4), where high-resolution inputs are used. We apologize for the confusion and will clarify it in our revision: "... the scalability issue w.r.t. the input resolution for vision transformers".
>
> ---
>
> W2: Clarify the novelty and contribution in comparison with existing clustering attention works.
>
> * As an integration scheme of attention and convolution, GLMix distinguishes itself from existing works in using Convs and MHSAs at different granularity levels, i.e., Convs on fine-grained feature maps and MHSAs on coarse semantic slots. We have found that with Convs applied on the feature grid, MHSAs can be aggressively applied to a few semantic slots while achieving comparable and even better performances than existing state-of-the-arts.
> * Unlike ClusTR [1] and TCFormer [4], which use DPC-KNN for the clustering, the soft clustering module in our work is fully learnable and does not rely on predefined rules.  In comparison with ClusterFormer [3] and PaCaViT [4], which perform cross-attention between the feature grid and cluster representations/slots, our work performs self-attention over the slots (i.e., queries and key-value pairs are both from the slots), making the attention even more lightweight. We will incorporate this discussion in the Related Work section in our revision.

---

### Author Rebuttal · Authors · 2024-08-07

We thank all reviewers for their constructive comments and suggestions. We are glad to see that reviewers consider our work novel/interesting/innovative (Reviewers fhiH/CazW/txiT), appreciate the semantic grouping effect brought by our design (Reviewers fhiH/CazW/txiT), and highlight the extensive experiments (Reviewers CazW/txiT). We address each reviewer's main concerns separately below.

---

### Decision · Program_Chairs · 2024-09-25

**Decision:**

Accept (poster)

**Comment:**

This paper presents a new method for integrating the benefits of Convs for local features and MHSA for global features, which is integrated using soft clustering. The novelty is well-recognized by the reviewers. The reviewers raised concerns about clarity, additional visualization, sensitivity to the choice of clustering strategy, etc. The author's rebuttal addressed most of the reviewers' concerns. Reviewer CazW suggests that most of his/her concerns have been addressed in the rebuttal and do not provide further feedback or rating updates. The meta-reviewer appreciates the additional observation that the authors' additional experiments suggest, which well explains the remaining issues by reviewer CazW.